# QUANTUM ARCHITECTURE SEARCH WITH UNSUPERVISED REPRESENTATION LEARNING

## ABSTRACT

Unsupervised representation learning presents new opportunities for advancing Quantum Architecture Search (QAS) on Noisy Intermediate-Scale Quantum (NISQ) devices. QAS is designed to optimize quantum circuits for Variational Quantum Algorithms (VQAs). Most QAS algorithms tightly couple the search space and search algorithm, typically requiring the evaluation of numerous quantum circuits, resulting in high computational costs and limiting scalability to larger quantum circuits. Predictor-based QAS algorithms mitigate this issue by estimating circuit performance based on structure or embedding. However, these methods often demand time-intensive labeling to optimize gate parameters across many circuits, which is crucial for training accurate predictors. Inspired by the classical neural architecture search algorithm *Arch2vec*, we investigate the potential of unsupervised representation learning for QAS without relying on predictors. Our framework decouples unsupervised architecture representation learning from the search process, enabling the learned representations to be applied across various downstream tasks. Additionally, it integrates an improved quantum circuit graph encoding scheme, addressing the limitations of existing representations and enhancing search efficiency. This predictor-free approach removes the need for large labeled datasets. During the search, we employ REINFORCE and Bayesian Optimization to explore the latent representation space and compare their performance against baseline methods. Our results demonstrate that the framework efficiently identifies high-performing quantum circuits with fewer search iterations.

## 1 INTRODUCTION

Quantum Computing (QC) has made significant progress over the past decades. Advances in quantum hardware and new quantum algorithms have demonstrated potential advantages (Stein et al., 2023) over classical computers in various tasks, such as image processing (Wang et al., 2022), reinforcement learning (Skolik et al., 2022), knowledge graph embedding (Ma et al., 2019), and network architecture search (Zhang et al., 2022; Giovagnoli et al., 2023; Du et al., 2022). However, the scale of quantum computers is still limited by environmental noise, which leads to unstable performance. These noisy intermediate-scale quantum (NISQ) devices lack fault tolerance, which is not expected to be achieved in the near future (Preskill, 2018). The variational quantum algorithm (VQA), a hybrid quantum algorithm that utilizes quantum operations with adjustable parameters, is considered a leading strategy in the NISQ era (Cerezo et al., 2021). In VQA, the parameterized quantum circuit (PQC) with trainable parameters is viewed as a general paradigm of quantum neural networks and has achieved notable success in quantum machine learning. These parameters control quantum circuit operations, adjusting the distribution of circuit output states, and are updated by a classical optimizer based on a task-specific objective function. Although VQA faces challenges such as Barren Plateaus (BP) and scalability issues, it has demonstrated the potential to improve performance across various domains, including image processing, combinatorial optimization, chemistry, and physics (Pramanik et al., 2022; Amaro et al., 2022; Tilly et al., 2022). One example of a VQA is the variational quantum eigensolver (VQE) (Peruzzo et al., 2014; Tilly et al., 2022), which approximates the ground state and offers flexibility for quantum machine learning. We are considering using VQE to evaluate the performance of certain quantum circuits.

Unsupervised representation learning seeks to discover hidden patterns or structures within unlabeled data, a well-studied problem in computer vision research (Radford et al., 2015). One common

approach is the autoencoder, which is effective for feature representation. It consists of an encoder and decoder, which first maps images into a compact feature space and then decodes them to reconstruct similar images. Beyond images, autoencoders can also learn useful features from graphs, such as encoding and reconstructing directed acyclic graphs (DAGs) or neural network architectures (Yan et al., 2020; Zhang et al., 2019; Pan et al., 2018; Wang et al., 2016). In most research, architecture search and representation learning are coupled, which results in inefficient searches heavily dependent on labeled architectures that require numerous evaluations. The *Arch2vec* framework aims to decouple representation learning from architecture search, allowing downstream search algorithms to operate independently (Yan et al., 2020). This decoupling leads to a smooth latent space that benefits various search algorithms without requiring extensive labeling.

Quantum architecture search (QAS) or quantum circuit architecture search is a framework for designing quantum circuits efficiently and automatically, aiming to optimize circuit performance (Du et al., 2022). Various algorithms have been proposed for QAS (Zhang et al., 2022; Du et al., 2022; Zhang et al., 2021; He et al., 2023a; Giovagnoli et al., 2023). However, most algorithms combine the search space and search algorithm, leading to inefficiency and high evaluation costs. The effectiveness of the search algorithm often depends on how well the search space is defined, embedded, and learned. Finding a suitable circuit typically requires evaluating different architectures many times. Although predictor-based QAS He et al. (2023a) can separate representation learning from the search algorithm, it often relies on labeling different architectures via evaluation, and the training performance depends heavily on the quantity and quality of evaluations and the embedding. In this work, we are inspired by the idea of decoupling, and we aim to conduct QAS without labeling. We seek to explore whether decoupling can embed quantum circuit architectures into a smooth latent space, benefiting predictor-free QAS algorithms.We summarise our contributions as follows:

- We have successfully incorporated decoupling into unsupervised architecture representation learning within QAS, significantly improving search efficiency and scalability. By applying REINFORCE and Bayesian optimization directly to the latent representation, we eliminate the need for a predictor trained on large labeled datasets, thereby reducing prediction uncertainty.
- Our proposed quantum circuit encoding scheme overcomes limitations in existing representations, enhancing search performance by providing more accurate and effective embeddings.
- Extensive experiments on quantum machine learning tasks, including quantum state preparation, max-cut, and quantum chemistry (Liang et al., 2019; Poljak & Rendl, 1995; Tilly et al., 2022), confirm the effectiveness of our framework. The pre-trained quantum architecture embeddings significantly enhance QAS across these applications.

## 2 RELATED WORK

**Unsupervised Graph Representation Learning.** Graph data is becoming a crucial tool for understanding complex interactions between real-world entities, such as biochemical molecules (Jiang et al., 2021), social networks (Shen et al., 2023), purchase networks from e-commerce platforms (Li et al., 2021), and academic collaboration networks (Newman, 2001). Graphs are typically represented as discrete data structures, making it challenging to solve downstream tasks due to large search spaces. Our work focuses on unsupervised graph representation learning, which seeks to embed graphs into low-dimensional, compact, and continuous representations without supervision while preserving the topological structure and node attributes. In this domain, approaches such as those proposed by Perozzi et al. (2014); Wang et al. (2016); Grover & Leskovec (2016); Tang et al. (2015) use local random walk statistics or matrix factorization-based objectives to learn graph representations. Alternatively, methods like Kipf & Welling (2016); Hamilton et al. (2017) reconstruct the graph's adjacency matrix by predicting edge existence, while others, such as Veličković et al. (2018); Sun et al. (2019); Peng et al. (2020), maximize the mutual information between local node representations and pooled graph representations. Additionally, Xu et al. (2019) investigate the expressiveness of Graph Neural Networks (GNNs) in distinguishing between different graphs and introduce Graph Isomorphism Networks (GINs), which are shown to be as powerful as the Weisfeiler-Lehman test (Leman & Weisfeiler, 1968) for graph isomorphism. Inspired by the success of *Arch2vec* (Yan et al., 2020), which employs unsupervised graph representation learning for

classical neural architecture search (NAS), we adopt GINs to injectively encode quantum architecture structures, as quantum circuit architectures can also be represented as DAGs.

**Quantum Architecture Search (QAS).** As discussed in the previous section, PQCs are essential as ansatz for various VQAs (Benedetti et al., 2019). The expressive power and entangling capacity of PQCs play a crucial role in their optimization performance (Sim et al., 2019). Poorly designed ansatz can suffer from limited expressive power or entangling capacity, making it difficult to reach the global minimum for an optimization problem. Moreover, such ansatz may be more prone to noise (Stilck França & Garcia-Patron, 2021), inefficiently utilize quantum resources, or lead to barren plateaus that hinder the optimization process (McClean et al., 2018; Wang et al., 2021). To address these challenges, QAS has been proposed as a systematic approach to identify optimal PQCs. The goal of QAS is to efficiently and effectively search for high-performance quantum circuits tailored to specific problems, minimizing the loss functions while adhering to constraints such as hardware qubit connections, native quantum gate sets, quantum noise models, training loss landscapes, and other practical considerations. Quantum architectures share many properties with neural network architectures, such as hierarchical, directed, and acyclic structures. As a result, QAS methods have been heavily inspired by techniques from NAS. Specifically, approaches such as greedy algorithms (Mitarai et al., 2018; Tang et al., 2021), evolutionary or genetic methods (Zhang & Zhao, 2022; Ding & Spector, 2022), RL-based engines (Kuo et al., 2021; Ostaszewski et al., 2021), Bayesian optimization (Duong et al., 2022), and gradient-based methods (Zhang et al., 2022) have all been employed to discover improved PQCs for VQAs. However, these methods require the evaluation of numerous quantum circuits, which is both time-consuming and computationally expensive. To mitigate this issue, predictor-based approaches (Zhang et al., 2021; He et al., 2023b) have been introduced, but they also face limitations. These approaches rely on large sets of labeled circuits to train predictors with generalized capabilities and introduce additional uncertainty into the search process, necessitating the reevaluation of candidate circuits. In this work, we propose a framework aimed at further addressing these challenges.

## 3 QAS WITH UNSUPERVISED REPRESENTATION LEARNING

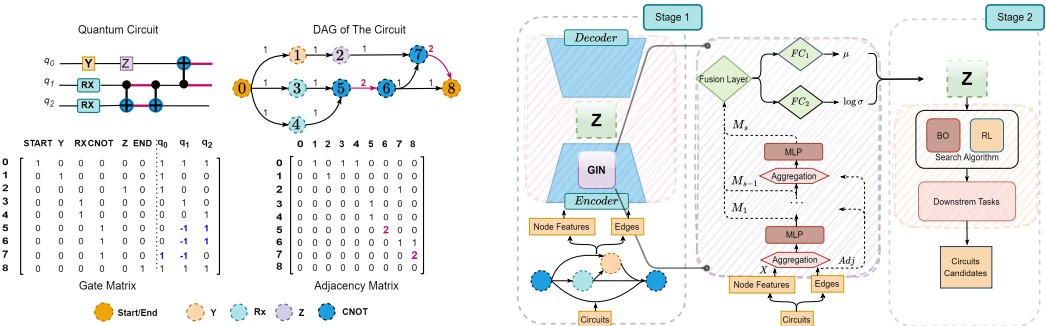

(a) Architecture encoding scheme      (b) Representation learning and search process

Figure 1: Illustration of our algorithm. In Figure 1a, each circuit's architecture is first transformed into a DAG and represented by two matrices. Each row of the gate matrix corresponds to a node in the graph, with one-hot encoding used to indicate the node type, and additional columns encoding position information, such as the qubits the gate acts on. For two-qubit gates, $-1$ and $1$ represent the control and target qubits, respectively. The weights in the adjacency matrix reflect the number of qubits involved in each interaction. In Figure 1b, the left side depicts the process of representation learning, where $Z$ represents the latent space of circuit architectures. In the middle, the encoder is shown as the mechanism used to learn this latent space. On the right, Bayesian optimization (BO) and reinforcement learning (RL) are employed to explore the latent space for various quantum machine learning tasks. The algorithm ultimately outputs a set of candidate circuits.

In this work, we present our method, as illustrated in Figure 1, which consists of two independent learning components: an autoencoder for circuit architecture representation learning, and a search process that includes both search and evaluation strategies. The search space is defined

by the number of gates in a circuit and an operation pool comprising general gate types such as `X`, `Y`, `Z`, `H`, `Rx`, `Ry`, `Rz`, `U3`, `CNOT`, `CY`, `CZ`. A random generator creates a set of circuit architectures based on predefined parameters, including the number of qubits, the number of gates, and the maximum circuit depth. These architectures are then encoded into two matrices and input into the autoencoder. The autoencoder independently learns a latent distribution from the search space and produces pre-trained architecture embeddings for the search algorithms. The evaluation strategy takes the circuit architectures generated by the search algorithm and returns a performance assessment. For evaluating circuit architectures, we use the ground state of a Hamiltonian for max-cut and quantum chemistry problems, and fidelity for quantum state preparation tasks.

## 3.1 CIRCUIT ENCODING SCHEME

We represent quantum circuits as DAGs using the circuit encoding scheme $\mathcal{E}^{GSQAS}$, as described in He et al. (2023b;a). Each circuit is transformed into a DAG by mapping the gates on each qubit to a sequence of nodes, with two additional nodes added to indicate the input and output of circuits. The resulting DAG is described by an adjacency matrix, as shown in Figure 1a. The set of nodes is further characterized by a gate matrix, which shows the node features including position information.

However, the encoding scheme $\mathcal{E}^{GSQAS}$ represents all occupied qubits as 1 without distinguishing between the control and target positions of two-qubit gates, which limits the effectiveness of circuit representation learning and leads to confusion during circuit reconstruction. Additionally, the adjacency matrix weights do not accurately reflect the original gate connections. To address these limitations, we propose a new encoding scheme. In our method, we explicitly encode positional information for two-qubit gates, such as `CNOT` and `CZ`, by assigning $-1$ to the control qubit and $1$ to the target qubit. Furthermore, we represent the number of qubits involved in an edge as the connection weights in the adjacency matrix, as shown in Figure 1a. These modifications enhance circuit representation learning and improve the overall effectiveness of the search.

## 3.2 VARIATIONAL GRAPH ISOMORPHISM AUTOENCODER

### 3.2.1 PRELIMINARIES

The most common graph autoencoders (GAEs) consist of an encoder and a decoder, where the encoder maps a graph into a feature space, and the decoder reconstructs the graph from those features. One prominent example is the variational graph autoencoder (VGAE), a promising framework for unsupervised graph representation learning that utilizes a graph convolutional network as its encoder and a simple inner product as its decoder (Kipf & Welling, 2016). In this work, however, we do not employ the common VGAE as a framework for learning latent representations. Instead, we utilize a more powerful encoder GIN (Xu et al., 2019).

**Definition 1.** *We are given a circuit created by $m$ gate types, $h$ gates and $g$ qubits. Then, the circuit can be described by a DAG $G = \{V, E\}$ with $n = h + 2 = |V|$ gate nodes including START and END. The adjacency matrix of graph $G$ is summarized in $n \times n$ matrix $A$ and its gate matrix $X$ is in size of $n \times (m + 2 + g)$. We further introduce $d$-dimensional latent variables $z_i$ composing latent matrix $Z = \{z_1, .., z_K\}^T$.*

### 3.2.2 ENCODER

The encoder GIN maps the structure and node features to latent representations $Z$. An approximation of the posterior distribution $q(Z|X, A)$ is:

$$q(Z|X, A) = \prod_{i=1}^{K} q(z_i|X, A), \tag{1}$$

where $q(z_i|X, A) = \mathcal{N}(z_i|\mu_i, \text{diag}(\sigma_i^2))$. The $L$-layer GIN generates the embedding matrix $M^{(s)}$ for $s$-layer by:

$$M^{(s)} = MLP^{(s)}((1 + \epsilon^{(s)}) \cdot M^{(s-1)} + \hat{A}M^{(s-1)}), s = 1, 2, ..., L, \tag{2}$$

Where $M^{(0)} = X$, and $\epsilon^{(s)}$ is a bias with a standard normal distribution for each layer. The $MLP$ is a multi-layer perceptron consisting of Linear-BatchNorm-LeakyReLU layers, and $\hat{A} = A + A^T$

transforms a directed graph into an undirected one to capture bi-directional information. In this work, we introduce a new fusion layer, a fully connected layer that aggregates feature information from all GIN layers, rather than just the last one. The mean $\mu = \text{GIN}_\mu(X, \hat{A}) = FC_1(M^{(L)})$ is computed using the fully connected layer $FC_1$, and similarly, the standard deviation $\sigma$ is computed via $FC_2$. We can then sample the latent matrix $Z \sim q(Z|X, A)$ by $z_i = \mu_i + \sigma_i \cdot \epsilon_i$. For all experiments, we use $L = 5$ GIN layers, a 16-dimensional latent vector $z_i$, and a GIN encoder with hidden sizes of 128. More details on the hyperparameters can be found in Appendix A.3.

### 3.2.3 DECODER

The decoder takes the sampled latent variables $Z$ as input to reconstruct both the adjacency matrix $A$ and the gate matrix $X = [X^t, X^q]$, where $X^t$ encodes the gate types and $X^q$ encodes the qubits on which the gates act. The generative process is summarized as follows:

$$p(A|Z) = \prod_{i=1}^{K} \prod_{j=1}^{K} p(A_{ij}|z_i, z_j), \text{ with } p(A_{ij}|z_i, z_j) = \text{ReLU}_j(F_1(z_i^T z_j)), \tag{3}$$

$$p(X|Z) = \prod_{i=1}^{K} p(x_i|z_i), \text{ with } p(x_i^t|z_i) = \text{softmax}(F_2(z_i)), p(x_i^q|z_i) = \tanh(F_2(z_i)), \tag{4}$$

where both $F_1$ and $F_2$ are trainable linear functions.

### 3.2.4 OBJECTIVE FUNCTION

The weights in the encoder and decoder are optimized by maximizing the evidence lower bound (ELBO) $\mathcal{L}$, which is defined as:

$$\mathcal{L} = E_{q(Z|X,A)}[\log p(X^{\text{type}}, X^{\text{qubit}}, A|Z)] - \text{KL}[(q(Z|X, A))||p(Z)], \tag{5}$$

where $\text{KL}[q(\cdot)||p(\cdot)]$ represents the Kullback-Leibler (KL) divergence between $q(\cdot)$ and $p(\cdot)$. We further adopt a Gaussian prior $p(Z) = \prod_i \mathcal{N}(z_i|0, I)$. The weights are optimized using minibatch gradient descent, with a batch size of 32.

## 3.3 ARCHITECTURE SEARCH STRATEGIES

### 3.3.1 REINFORCEMENT LEARNING (RL)

After conducting initial trials with PPO (Schulman et al., 2017) and A2C (Huang et al., 2022), we adopt REINFORCE (Williams, 1992) as a more effective reinforcement learning algorithm for architecture search. In this approach, the environment's state space consists of pre-trained embeddings, and the agent uses a one-cell LSTM as its policy network. The agent selects an action, corresponding to a sampled latent vector based on the distribution of the current state, and transitions to the next state based on the chosen action. The reward for max-cut and quantum chemistry tasks is defined as the ratio of energy to ground energy, with values outside the range [0, 1] clipped to 0 or 1. For the state preparation task, circuit fidelity is used as the reward. We employ an adaptive batch size, with the number of steps per training epoch determined by the average reward of the previous epoch. Additionally, we use a linear adaptive baseline, defined by the formula $B = \alpha \cdot B + (1 - \alpha) \cdot R_{avg}$, where $B$ denotes the baseline, $\alpha$ is a predefined value in the range [0,1], and $R_{avg}$ is the average reward. Each run in this work involves 1000 searches.

### 3.3.2 BAYESIAN OPTIMIZATION (BO)

As another search strategy used in this work without labeling, we employ Deep Networks for Global Optimization (DNGO)(Snoek et al., 2015) in the context of BO. We adopt a one-layer adaptive BO regression model with a basis function extracted from a feed-forward neural network, consisting of 128 units in the hidden layer, to model distributions over functions. Expected Improvement (EI)(Mockus, 1977) is selected as the acquisition function. EI identifies the top-k embeddings for each training epoch, with a default objective value of 0.9. The training begins with an initial set of 16 samples, and in each subsequent epoch, the top-k architectures proposed by EI are added to the batch. The network is retrained for 100 epochs using the architectures from the updated batch. This process is iterated until the predefined number of search iterations is reached.

## 4 EXPERIMENTAL RESULTS

To demonstrate the effectiveness and generalization capability of our approach, we conduct experiments on three well-known quantum computing applications: quantum state preparation, max-cut, and quantum chemistry. For each application, we start with a simple example involving 4 qubits and then progress to a more complex example with 8 qubits. We utilize a random generator to create 100,000 circuits as the search space, and all experiments are performed on a noise-free simulator during the search process. Detailed settings are provided in Appendix A.2. We begin by evaluating the model's pre-training performance for unsupervised representation learning (§4.1), followed by an assessment of QAS performance based on the pre-trained latent representations (§4.2).

### 4.1 PRE-TRAINING PERFORMANCE

**Observation (1):** GAE and VGAE (Kipf & Welling, 2016) are two popular baselines for NAS. In an attempt to find models capable of capturing superior latent representations of quantum circuit architectures, we initially applied these two well-known models. However, due to the increased complexity of quantum circuit architectures compared to neural network architectures, these models failed to deliver the expected results. In contrast, models based on GINs (Xu et al., 2019) successfully obtained valid latent representations, attributed to their more effective neighbor aggregation scheme. Table 1 presents a performance comparison between the original model using the $\mathcal{E}^{GSQAS}$ encoding and the improved model with our enhanced encoding for 4, 8, and 12 qubit circuits, evaluated across five metrics: Accuracy$ops$, which measures the reconstruction accuracy of gate types in the gate matrix for the held-out test set; Accuracy$qubit$, which reflects the reconstruction accuracy of qubits that the gates act on; Accuracy$adj$, which measures the reconstruction accuracy of the adjacency matrix; Falpos$mean$, which represents the mean false positives in the reconstructed adjacency matrix; and KLD (KL divergence), which indicates the continuity and smoothness of the latent representation. The results in the table indicate that the improved model with our enhanced encoding achieves comparable or better than the original. This improvement can be attributed to two factors: first, the new encoding better captures the specific characteristics of the circuits, and second, the fusion of outputs from multiple layers of GIN helps retain shallow information, resulting in more stable training.

| Qubit | Model | Metric | | | | |
|---|---|---|---|---|---|---|
| | | Accuracy$_{ops}$ | Accuracy$_{qubit}$ | Accuracy$_{adj}$ | Falpos$_{mean}$ | KLD |
| 4 | GSQAS | 99.99 | 99.99 | 99.91 | 100.00 | 0.061 |
| 4 | Ours | **100** | 99.97 | 98.89 | **23.41** | **0.045** |
| 8 | GSQAS | 86.69 | 99.98 | 99.82 | 100.00 | 0.038 |
| 8 | Ours | **100** | 98.65 | 97.34 | **7.35** | **0.029** |
| 12 | GSQAS | 86.69 | 99.94 | 99.70 | 100.00 | 0.028 |
| 12 | Ours | **98.67** | 99.14 | 97.79 | **4.75** | **0.022** |

Table 1: Pretraining model performance of 4-, 8-, and 12-qubit circuits across the four metrics.

**Observation (2):** In Figure 2, we employ two popular techniques, PCA (Shlens, 2014) and t-SNE (Van der Maaten & Hinton, 2008), to visualize the high-dimensional latent representations of 4- and 12-qubit quantum machine learning (QML) applications based on our pre-trained models. The results highlight the effectiveness of our new encoding approach for unsupervised clustering and high-dimensional data visualization. The figures show that the latent representation space of quantum circuits is smooth and compact, with architectures of similar performance clustering together when the search space is limited to 4 qubits. Notably, high-performance quantum circuit architectures are concentrated on the right side of the visualizations. In particular, PCA yields exceptionally smooth and compact representations with strong clustering effects, making it easier and more efficient to conduct QAS within such a structured latent space. This provides a robust foundation for our QAS algorithms.

For the 12-qubit latent space, high-performance circuits (shown in red) are less prominent, likely due to the fact that the 100,000 circuit structures represent only a finite subset of the possibilities for 12-qubit circuit. As a result, the number of circuits that can be learned is limited. Most high-

performance circuits are distributed along the left edge of the latent space, with a color gradient transitioning from dark to light as one moves from right to left.

Compared with subfigures 2i, 2j, 2k, 2l, 2m, and 2k, which utilize the encoding scheme $\mathcal{E}^{GSQAS}$ and show more loosely distributed red points, our new encoding results in a more concentrated and smoother latent representation, as demonstrated in subfigures 2a, 2b and 2c.

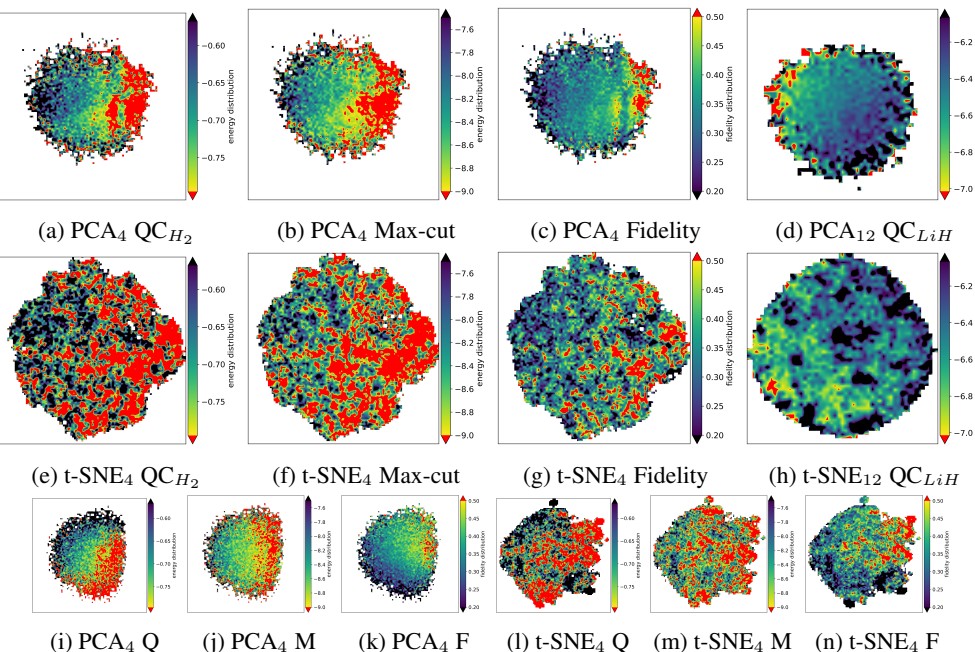

Figure 2: The 2D smooth visualizations of the latent representations for the 4- and 12-qubit cases, using PCA and t-SNE. The color encoding reflects the achieved energy of 100,000 randomly generated circuits. These latent representations are introduced for three QML tasks: Quantum Chemistry, Max-cut, and fidelity. The graphs illustrate the energy or fidelity distribution of the circuits, where red denotes circuits with an energy lower than $-0.80/-0.90/-7.01$, Ha or a fidelity higher than 0.5. The subfigures in the first two rows display the results of our model with KL divergence, while the subfigures at the bottom visualize the 4-qubit latent space using the existing encoding scheme $\mathcal{E}^{GSQAS}$.

## 4.2 QUANTUM ARCHITECTURE SEARCH (QAS) PERFORMANCE

**Observation (1):** In Figure 3, we present the average reward per 100 searches for each experiment. The results show that both the REINFORCE and BO methods effectively learn to navigate the latent representation, leading to noticeable improvements in average reward during the early stages. In contrast, Random Search fails to achieve similar improvements. Furthermore, although the plots indicate slightly higher variance in the average reward for the REINFORCE and BO methods compared to Random Search, their overall average reward is significantly higher than that of Random Search.

**Observation (2):** In Figure 4, we illustrate the number of candidate circuits found to achieve a preset threshold after performing 1000 searches using the three search methods. The results show that the 8-qubit experiments are more complex, resulting in fewer circuits meeting the requirements within the search space. Additionally, within a limited number of search iterations, both the REINFORCE and BO methods are able to discover a greater number of candidate circuits that meet the threshold, even in the worst case, i.e., when comparing the minimal number of candidates. Notably, their performance significantly surpasses that of the Random Search method, especially REINFORCE, despite the fact that the difference between the minimal and maximal number of candidates indicates that REINFORCE is more sensitive to the initial conditions compared to the other two approaches. These findings highlight the clear improvements and advantages introduced by QAS based on the

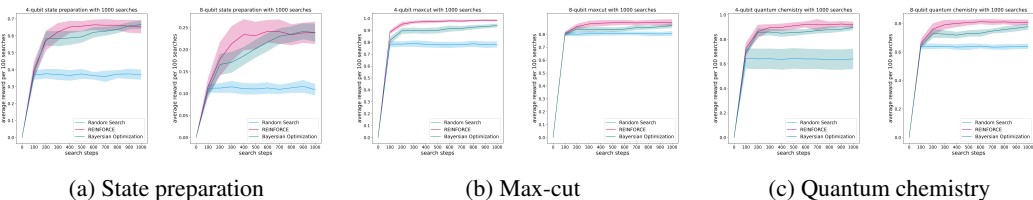

(a) State preparation      (b) Max-cut      (c) Quantum chemistry

Figure 3: Average rewards from the six sets of experiments. In subfigures (a), (b), and (c), the left panels show results from the 4-qubit experiments, while the right panels show results from the 8-qubit experiments. Each plot presents the average reward across 50 independent runs (each with different random seeds) given 1000 search queries. The shaded areas in the plots represent the standard deviation of the average rewards.

latent representation, which enables the efficient discovery of numerous high-performance candidate circuits while reducing the number of searches required.

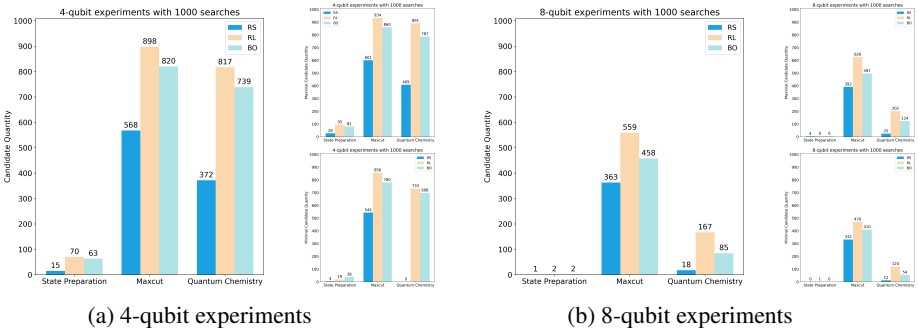

(a) 4-qubit experiments          (b) 8-qubit experiments

Figure 4: The candidate quantities for the 4-qubit and 8-qubit applications. RS, RL, and BO refer to Random Search, REINFORCE, and Bayesian Optimization, respectively. The reward threshold for all 4-qubit experiments is 0.95, while for the more complex 8-qubit experiments, the thresholds are softer: 0.75 for state preparation, 0.925 for max-cut, and 0.95 for quantum chemistry. Each experiment is performed with 1000 queries, meaning only 1000 samples are drawn from a search space of 100,000 circuits. Additionally, the left-hand side of subfigures (a) and (b) shows the average results over 50 runs (with different random seeds), while the right-hand side shows the maximum and minimum candidate quantities across the 50 runs.

**Observation (3):** In Table 2, we compare various QAS methods with our approach on the 4-qubit state preparation task, using a circuit space of 100,000 circuits and limiting the search to 1000 queries. $GNN^{URL}$ and $GSQAS^{URL}$ represent predictor-based methods from He et al. (2023b) and He et al. (2023a), respectively, both employing our pre-trained model. $QAS^{URL}_{RL(BO)}$ denotes the QAS approach with REINFORCE (BO) used in this work. The average results over 50 runs indicate that both the predictor-based methods and our approach are capable of identifying a significant number of high-performance circuits with fewer samples. However, predictor-based methods rely on labeled circuits to train predictors, introducing uncertainty as they may inadvertently filter out well-performing architectures along with poor ones. While a higher $F_{thr}$ value filters out more low-performance circuits, increasing the proportion of good architectures in the filtered space, it also sacrifices many well-performing circuits, which can lead to improved Random Search performance but at the cost of excluding some optimal circuits. Despite these trade-offs, our method achieves comparable performance to predictor-based methods, demonstrating higher efficiency in terms of $N_{QAS}/N_{eval}$ while requiring fewer circuit evaluations. In Appendix A.4, we present the best candidate circuits acquired by each of the three methods for every experiment.

**Observation (4):** In Table 3, we present the search performance across different frameworks and encoding methods, focusing on 4-, 8-, and 12-qubit quantum chemistry tasks for comparison. In most cases, our encoding method achieves the highest search efficiency, although the performance

| Method | $Task$ | $F_{thr}$ | $N_{lbl}$ | $N_{rest}$ | $N_{>0.95}$ | $N_{eval}$ | $N_{QAS}$ | $N_{QAS}/N_{eval}$ |
|---|---|---|---|---|---|---|---|---|
| $GNN^{URL}$ | Fidelity | 0.5 | 1000 | 21683 | 780 | 2000 | 36 | 0.0180 |
| | Max-Cut | 0.9 | 1000 | 45960 | 35967 | 2000 | 783 | 0.3915 |
| | QC-4$_{H_2}$ | 0.8 | 1000 | 65598 | 18476 | 2000 | 278 | 0.1390 |
| $GSQAS^{URL}$ | Fidelity | 0.5 | 1000 | 21014 | 768 | 2000 | 37 | 0.0185 |
| | Max-Cut | 0.9 | 1000 | 43027 | 33686 | 2000 | 785 | 0.3925 |
| | QC-4$_{H_2}$ | 0.8 | 1000 | 30269 | 19889 | 2000 | 658 | 0.3290 |
| Random Search | Fidelity | - | 0 | 100000 | 1606 | 1000 | 15 | 0.0150 |
| | Max-Cut | - | 0 | 100000 | 57116 | 1000 | 568 | 0.5680 |
| | QC-4$_{H_2}$ | - | 0 | 100000 | 37799 | 1000 | 371 | 0.3710 |
| $QAS^{URL}_{RL(BO)}$ | Fidelity | - | 0 | 100000 | **1606** | **1000** | **69**(63) | **0.0690**(0.0630) |
| | Max-Cut | - | 0 | 100000 | **57116** | **1000** | **898**(820) | **0.8980**(0.8200) |
| | QC-4$_{H_2}$ | - | 0 | 100000 | **37799** | **1000** | **817**(739) | **0.8170**(0.7390) |

Table 2: Compare the QAS performance of different QAS methods for the 4-qubit tasks. URL denotes unsupervised representation learning, $F_{thr}$ is the threshold to filter poor-performance architectures, $N_{lbl}$, $N_{rest}$ and $N_{>0.95}$ refer to the number of required labeled circuits, rest circuits after filtering and the circuits that achieve the performance higher than 0.95 in the rest circuits respectively. $N_{eval}$ represents the number of evaluated circuits, i.e. the sum of the number of labeled and sampled circuits, $N_{QAS}$ is the number of searched candidates in average of 50 runs.

| Method | Encoding $\mathcal{E}$ | $N_{rest}$ | $N_{eval}$ | $N_{QAS}$ | $N_{QAS}/N_{eval}$ |
|---|---|---|---|---|---|
| $GSQAS_4$ | GSQAS | 25996 | 2000 | 625 | 0.3125 |
| | Ours | 30269 | 2000 | **658** | **0.3290** |
| $GSQAS_{12}$ | GSQAS | 60088 | 2000 | **283** | **0.1415** |
| | Ours | 60565 | 2000 | 276 | 0.1380 |
| $QAS_{RL-4}$ | GSQAS | 100000 | 1000 | 760 | 0.7600 |
| | Ours | 100000 | 1000 | **817** | **0.8170** |
| $QAS_{RL-8}$ | GSQAS | 100000 | 1000 | 160 | 0.1600 |
| | Ours | 100000 | 1000 | **167** | **0.1670** |
| $QAS_{RL-12}$ | GSQAS | 100000 | 1000 | **422** | **0.4220** |
| | Ours | 100000 | 1000 | 392 | 0.3920 |

Table 3: We compare the QAS performance of different encodings using various search methods. For the 4- and 12-qubit quantum chemistry tasks, we select $H_2$ and $LiH$, respectively, while for the 8-qubit task, we use the TFIM. The results represent the average of 50 runs.

for the 12-qubit task is slightly lower than with another encoding method. Combined with the representation learning results in Figure 2, we observe that the search is significantly more efficient when the learned circuit representation is smooth and concentrated. For the 12-qubit experiments, the circuits used for representation learning may be insufficient to fully capture the search space, leading to representation learning failures, as shown in Figure 2d, and resulting in a decline in search efficiency.

## 5 CONCLUSION

Inspired by the *Arch2vec* method (Yan et al., 2020), we focus on exploring whether unsupervised architecture representation learning can enhance QAS. By decoupling unsupervised architecture representation learning from the QAS process, we successfully eliminate the need for a large number of labeled circuits. Additionally, our proposed quantum circuit encoding scheme addresses limitations in existing representations, improving search performance through more accurate and effective embeddings. Furthermore, our framework conducts QAS without relying on a predictor by directly applying search algorithms, such as REINFORCE and Bayesian Optimization (BO), to the latent representations. We have demonstrated the effectiveness of this approach through various experiments. In our framework, the success of QAS depends on the quality of unsupervised architecture representation learning and the selection of search algorithms. Thus, we recommend further investigation into architecture representation learning for QAS, as well as the development of more efficient search strategies within the latent representation space.

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

# A APPENDIX

## A.1 CIRCUIT GENERATOR SETTINGS

The predefined operation pool which defines allowed gates in circuits is important for QAS as well, because a terrible operation pool such as one with no rotation gates or no control gates cannot generate numerous quantum circuits with excellent expressibility and entanglement capability. This makes the initial quantum search space poor, so it will influence our further pre-training and QAS process. Therefore, we choose some generally used quantum gates in PQCs as our operation pool {X, Y, Z, H, Rx, Ry, Rz, U3, CNOT, CZ, CY} for the circuit generator to guarantee the generality of our quantum circuit space. Other settings of the circuit generator are summarized below:

Table 4: Description of settings predefined for the circuit generator.

| Hyperparameter | Hyperparameter explanation | Value for 4/8/12-qubit experiments |
|---|---|---|
| num-qubits | the number of qubits | 4/8/12 |
| num-gates | the number of gates in a circuit | 10/20/30 |
| max-depth | the maximal depth in a circuit | 5 |
| num-circuits | required the number of circuits | $10^5$ |

## A.2 APPLICATION SETTINGS

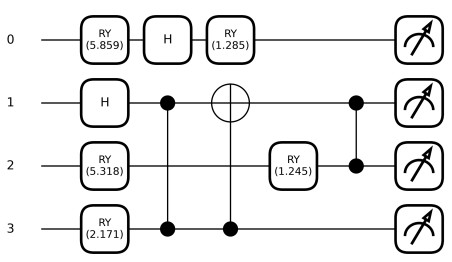

(a) The target circuit of the 4-qubit state preparation

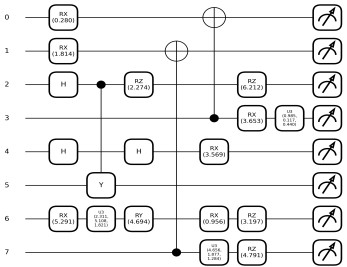

(b) The target circuit of the 8-qubit state preparation

Figure 5: The circuits used to generate the target states.

**Quantum State Preparation.** In quantum information theory, fidelity (Liang et al., 2019) is an important metric to measure the similarity of two quantum states. By introducing fidelity as the performance index, we aim to maximize the similarity of the final state density operator with a certain desired target state. We first obtain the target state by randomly generating a corresponding circuit, and then with a limited number of sample circuits, we use the search methods to search candidate circuits that can achieve a fidelity higher than a certain threshold. During the search process, the fidelity can be directly used as a normalized reward function since its range is [0, 1]. Figure 5 shows the circuits used to generate the corresponding target states.

**Max-cut Problems.** The max-cut problem (Poljak & Rendl, 1995) consists of finding a decomposition of a weighted undirected graph into two parts (not necessarily equal size) such that the sum of the weights on the edges between the parts is maximum. Over these years, the max-cut problem can be efficiently solved with quantum algorithms such as QAOA (Villalba-Diez et al., 2021) and VQE (using eigenvalues). In our work, we address the problem by deriving the Hamiltonian of the graph and using VQE to solve it. We use a simple graph with the ground state energy $-10\,Ha$ for the 4-qubit experiment and a relatively complex graph with the ground state energy $-52\,Ha$ in the case

of the 8-qubit experiment. Furthermore, we convert the energy into a normalized reward function integral to the search process. The visual representations of these graphs are presented below:

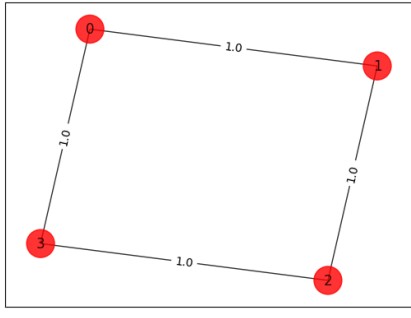
(a) The 4-qubit max-cut graph

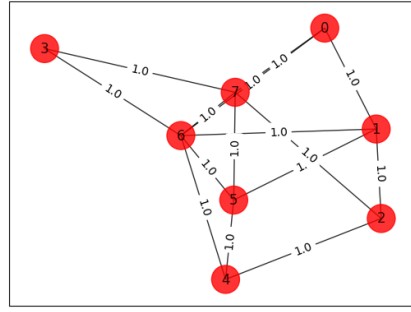
(b) The 8-qubit max-cut graph

Figure 6: The graphs of the experiments on max-cut problems.

**Quantum Chemistry.**   In the field of QC, VQE (Peruzzo et al., 2014; Tilly et al., 2022) is a hybrid quantum algorithm for quantum chemistry, quantum simulations, and optimization problems. It is used to compute the ground state energy of a Hamiltonian based on the variational principle. For the 4- and 12-qubit quantum chemistry experiment, we use the Hamiltonian of the molecule $H_2$ and $LiH$ and its common approximate ground state energy $-1.136\ Ha$ and $-7.88\ Ha$ as the optimal energy. As for the 8-qubit experiment, we consider $n = 8$ transverse field Ising model (TFIM) with the Hamiltonian as follows:

$$\boldsymbol{H} = \sum_{i=0}^{7} \sigma_z^i \sigma_z^{(i+1)\ mod\ 6} + \sigma_x^i. \tag{6}$$

We design some circuits to evaluate the ground state energy of the above Hamiltonian and get an approximate value $-10\ Ha$ as the optimal energy. According to the approximate ground state energy, we can use our methods to search candidate circuits that can achieve the energy reaching a specific threshold. In the process of searching for candidates, the energy is normalized as a reward function with the range [0, 1] to guarantee search stability.

A.3   HYPERPARAMETERS OF PRE-TRAINING

Table 5 shows the hyperparameter settings of the pre-training model for 4-qubit and 8-qubit experiments.

Table 5: Description of hyperparameters adopted for pre-training.

| Hyperparameter | Hyperparameter explanation | Value for 4/8/12-qubit experiments |
|---|---|---|
| bs | batch size | 32 |
| epochs | traning epochs | 16 |
| dropout | decoder implicit regularization | 0.1 |
| normalize | input normalization | True |
| input-dim | input dimension | 2+#gates+#qubits |
| hidden-dim | dimension of hidden layer | 128 |
| dim | dimension of latent space | 16 |
| hops | the number of GIN layers ($L$ in eq.2) | 5 |
| mlps | the number of MLP layers | 2 |

## A.4 BEST CANDIDATE CIRCUITS

**Observation (5):** In Appendix A.4, we present the best candidate circuits acquired by each of the three methods for every experiment. These circuits exhibit a higher likelihood of being discovered by REINFORCE and BO in contrast to Random Search. This observation underscores the superior search capabilities of REINFORCE and BO in navigating the large and diverse search space generated by our approach, which is based on a random generator derived from a fixed operation pool. Unlike conventional approaches that adhere to layer-wise circuit design baselines, our method excels in discovering circuits with fewer trainable parameters. This characteristic is of paramount importance when addressing real-world optimization challenges in QAS. In conclusion, our approach not only enhances the efficiency of candidate circuit discovery but also accommodates the distinct characteristics of various problem domains through a large and diverse search space.

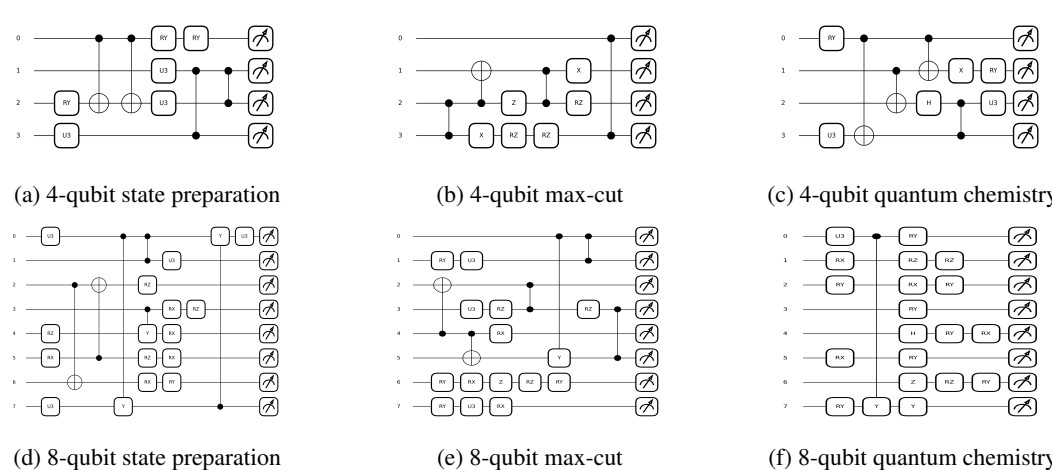

(a) 4-qubit state preparation     (b) 4-qubit max-cut     (c) 4-qubit quantum chemistry

(d) 8-qubit state preparation     (e) 8-qubit max-cut     (f) 8-qubit quantum chemistry

Figure 7: Best candidates of the six experiments in 50 runs.

