# OpenReview forum: "Quantum Architecture Search With Unsupervised Representation Learning"
_ICLR.cc/2025/Conference — ICLR 2025 Conference Withdrawn Submission_

### Official Review · Reviewer_RkLH · 2024-10-21

**Soundness:** 2
**Presentation:** 2
**Contribution:** 2
**Rating:** 3
**Confidence:** 4

**Summary:**

This paper presents an algorithm for quantum architecture search based on the arch2vec algorithm. Specifically, they generate quantum circuits and learn a latent representation for their structure using a graph based autoencoder, then optimize generated circuits based on the learned latent representation.

**Strengths:**

- The code is open source which is beneficial for reproducibility
- Figure 1 and the explanation is helpful
- The latent approach to QAS to avoid expensive repeated circuit evaluations during every search is interesting and warrants further research as an idea

**Weaknesses:**

- The empirical justifications are lacking. Given that circuit width and depth will be much larger than the circuits shown, strong arguments are necessary in regards to scalability. However, existing results seem to indicate pretty negative correlations with scale. Figures 3 and 4 show that 8 qubits achieves much lower performance than the 4 qubit. The scale is also limited when it comes to simulations, given that up to 20 qubits is doable on laptops. There is no compelling arguments for how these methods would perform under practically interesting/relevant conditions and this needs to be included.
- Subjective statements could be quantified: e.g. “show more loosely distributed red points, our new encoding results in a more concentrated and smoother latent representation” this could be analyzed rather than the just claimed visually
- Table 1 shows the results of the pretraining for the latent space, which would benefit from uncertainty estimates (since these are dependent, ideally minimally, on the initializations/random keys)
- It would be worth showing other algorithms as well (even if in the appendix) that are mentioned in the text. Specifically, PPO and A2C are mentioned as algorithms that were evaluated but didn’t perform as well as REINFORCE, having plots showing that in the appendix would be worthwhile
- Minor graphical points: e.g. coloring on figure 3(c) left is wrong for random search
- Minor grammatical points: e.g. “We are considering using” → “we use”

**Questions:**

- How scalable is the representation? For realistic circuits that could potentially offer some advantage, e.g. with hundreds of qubits and depth close to linear, are these graphs learnable?

---

### Official Review · Reviewer_XR1U · 2024-11-02

**Soundness:** 2
**Presentation:** 2
**Contribution:** 2
**Rating:** 5
**Confidence:** 3

**Summary:**

This paper, Quantum Architecture Search with Unsupervised Representation Learning, presents a novel approach to optimizing quantum circuit architectures for variational quantum algorithms on NISQ devices. It introduces a predictor-free Quantum Architecture Search (QAS) framework that uses unsupervised representation learning to eliminate the need for labeled data. The authors propose an improved encoding scheme for representing quantum circuits as directed acyclic graphs (DAGs) and use a variational graph isomorphism autoencoder (GIN) to learn smooth latent representations of circuit architectures. To explore this representation space, they apply reinforcement learning and Bayesian optimization, demonstrating the framework’s effectiveness across quantum machine learning tasks like quantum state preparation, max-cut, and quantum chemistry.

**Strengths:**

1.	The paper introduces a novel approach to Quantum Architecture Search (QAS) that leverages unsupervised representation learning, eliminating the need for labeled datasets and predictors.
2.	The authors propose an improved quantum circuit encoding scheme, representing circuits as directed acyclic graphs (DAGs) with specific positional information for two-qubit gates.
3. The paper demonstrates the versatility of its framework by employing both reinforcement learning (REINFORCE) and Bayesian optimization (BO) as search strategies within the learned latent space.

**Weaknesses:**

See the questions listed below.

**Questions:**

1. Could the authors specify the adaptations made in the encoder and decoder specifically for the QAS task? While the paper describes the overall model structure including the GINs and RLs, it would be helpful to clarify how these changes, such as the addition of a fusion layer after the GIN, influence the learning and final performance. Currently, the innovation in this area feels somewhat ambiguous, and further detail on how the QAS-related adaptation contributes to performance would be insightful.

2. The scalability of this approach is not fully demonstrated in the current experiments, as the main results are based on circuits with up to 12 qubits (mainly over 4 and 8 qubits) and a maximum depth of 5. It would be beneficial to include experiments on larger qubit circuits and deeper circuits to better assess the model’s scalability and robustness in more complex settings.

3. It appears that only two baselines are used for comparison. Expanding the number of baseline methods would provide a clearer picture of the model’s relative performance and robustness, enhancing the strength of the comparisons.

4. I suggest reorganizing the paper structure to enhance readability. For instance, consolidating hyperparameter values in the experimental section, rather than embedding them within the methodology, would create a more streamlined and accessible flow for readers.

5. In addition to 1, an ablation study of important modules may help the demonstration.

---

### Official Review · Reviewer_QH74 · 2024-11-03

**Soundness:** 2
**Presentation:** 3
**Contribution:** 2
**Rating:** 3
**Confidence:** 5

**Summary:**

The paper propose a framework for Quantum Architecture Search (QAS) utilizing unsupervised representation learning to optimize quantum circuits for Variational Quantum Algorithms (VQAs). The framework is inspired by the Arch2vec algorithm for classical neural architectures and it decouples representation learning from search, and enables efficient latent-space exploration without labeled datasets. It proposes an improved quantum circuit encoding scheme that refines gate connectivity representations and a variational graph isomorphism autoencoder (GIN) for encoding circuit structures. The QAS search process leverages REINFORCE-based reinforcement learning and Bayesian optimization (BO), and shows efficiency improvements in identifying high-performing circuits for quantum state preparation, max-cut problem, and quantum chemistry applications.

**Strengths:**

1. The paper tries to decouple the representation learning of the quantum circuit to the search process, which is an important direction towards more general search of variational quantum ansatz.
2. The paper introduces a refined encoding with novel gate matrix and adjacency matrix that gets the detailed circuit characteristics, control and target qubit positions in multi-qubit gates. This scheme benefits downstream tasks by enhancing the structural understanding to the circuit.

**Weaknesses:**

1. The efforts to find a way that decouples the circuit ansatz representation to the search process is important. However, the proposed method is not a scalable way to do it. For small size circuit, it is easy for the encoder and decoder to encode the circuit with a compact vector. However, the paper only demonstrated the number of qubit fewer than 12. and the reconstruction accuracy of 12 qubits is only 98.76%. The accuracy of reconstruction will degrades significantly when the number of qubit increases, as the design space increases exponentially.
2. Beside the difficulty to encode circuit with large number of qubit, the data required to train the encode and decoder is also increasing exponentially to cover the space.
3. For large number of qubits, to achieve arbitrary unitary, the required number of 2 q gate increases exponentially, the size of adjacent matrix and gate matrix also increase exponentially.
The author should justify the effectiveness and scalability to a reasonable size, for example, 50 qubits. Because a 12 qubit VQA will not generate any quantum advantage.

minor
1. Line 290, the "ops", "qubit", "adj", "mean" should be underscore
2. The text is too small to read in figure 4

**Questions:**

1. How to make sure in the decoder output, the generated adjacent matrix and the gate matrix match with each other.
2. Why use GAE and VGAE as baseline, they are pretty old models.
3. Theoretically, it is very hard to efficiently represent an arbitrary quantum circuit ansatz without a exponential scaling size of vector. Do you really need arbitrary ansatz? Some ansatz block like efficientSU2 are comprehensive enough to represent arbitrary unitary, why we don't use them as building blocks? Conceptually, it is similar case in classical NAS, we don't search for arbitrary neural architecture, instead, we search with building blocks such as conv, attention, MLP etc.

---

### Official Review · Reviewer_nFoZ · 2024-11-04

**Soundness:** 1
**Presentation:** 2
**Contribution:** 1
**Rating:** 3
**Confidence:** 5

**Summary:**

The authors focused on automatically designing the quantum circuit to reach acceptable accuracy while keeping the circuit at a very low circuit depth. They tried to utilize a representation learning model to evaluate the quantum circuits generated by Reinforcement Learning and Bayesian Optimization. Numerical results are provided on state preparation, Max-cut, and Ground-state energy estimation problems with the comparison to a AAAI paper and random search.

**Strengths:**

Well-structured paper.

**Weaknesses:**

1. First, the authors should pay more attention to the background check of the related works in the quantum architecture search area.  The conclusion in section 2 that "these methods require the evaluation of numerous quantum circuits" is indeed incorrect for the cited paper. Recent QAS approaches can generate circuit architecture while optimizing the internal parameters (you can refer to the survey [1]), which does not require evaluating the generated circuits.
2. The most important concern is the motivation. The paper is titled "Quantum Architecture Search...," but it is indeed about representation learning of a quantum circuit. The search methods proposed in this paper were not designed by the authors. The authors tried to utilize the learned representation to evaluate the generated quantum circuit, which I find quite strange. The representation model should be trained with a certain qubit number and a certain Hamiltonian, indicating the proposed method is not scalable or generalizable. If you adopt recent models on the QAS, you can generate the quantum ansatz with optimized parameters, which can be easily evaluated with a simple evaluation of the output state with the given Hamiltonian or target state. I don't see the need to utilize a representation learning model here.
3. The evaluation metric can not reflect the actual performance of the proposed methods. The ground-state estimation for quantum chemistry simulation requires admissible results within chemical accuracy (1.6 mHa) [3]. Generating a circuit with a maximum depth of 5 layers is too naive for the tasks. The only baseline method is from [2], which is weak considering the fact that there are numerous other QAS methods.







[1] Quantum circuit synthesis and compilation optimization: Overview and prospects

[2] Experimental quantum computational chemistry with optimized unitary coupled cluster ansatz

**Questions:**

No questions

---

### Note · Authors · 2024-11-27

**Comment:**

Dear ICLR Program Committee,

I hope this message finds you well. After carefully reviewing the feedback provided by the reviewers, I have decided to withdraw my submission, Submission Number 12432, from consideration. While I appreciate the time and effort the reviewers have put into evaluating my work and acknowledge that some of the comments are constructive and will contribute to improving the manuscript, I believe that the overall feedback does not align with the intent and contributions of the paper.

At this point, I believe it is best to further refine the work and address these points outside of the current review process.

Thank you for the opportunity to submit my work to ICLR.

Best regards,

**Withdrawal Confirmation:**

I have read and agree with the venue's withdrawal policy on behalf of myself and my co-authors.